# Health Effects of a 12-Week Web-Based Lifestyle Intervention for Physically Inactive and Overweight or Obese Adults: Study Protocol of Two Randomized Controlled Clinical Trials

**DOI:** 10.3390/ijerph19031393

**Published:** 2022-01-26

**Authors:** Judith Brame, Jan Kohl, Ramona Wurst, Reinhard Fuchs, Iris Tinsel, Phillip Maiwald, Urs Fichtner, Christoph Armbruster, Martina Bischoff, Erik Farin-Glattacker, Peter Lindinger, Rainer Bredenkamp, Albert Gollhofer, Daniel König

**Affiliations:** 1Department of Sport and Sport Science (DoSS), University of Freiburg, 79117 Freiburg, Germany; jan.kohl@sport.uni-freiburg.de (J.K.); ramona.wurst@sport.uni-freiburg.de (R.W.); reinhard.fuchs@sport.uni-freiburg.de (R.F.); ag@sport.uni-freiburg.de (A.G.); daniel.koenig@univie.ac.at (D.K.); 2Section of Health Care Research and Rehabilitation Research (SEVERA), Medical Center—University of Freiburg, Faculty of Medicine, University of Freiburg, 79106 Freiburg, Germany; iris.tinsel@uniklinik-freiburg.de (I.T.); phillip.maiwald@uniklinik-freiburg.de (P.M.); urs.fichtner@uniklinik-freiburg.de (U.F.); christoph.armbruster@uniklinik-freiburg.de (C.A.); martina.bischoff@uniklinik-freiburg.de (M.B.); erik.farin@uniklinik-freiburg.de (E.F.-G.); 3Scientific Working Group in Smoking Cessation (WAT) e.V., Department of Psychiatry and Psychotherapy, University Hospital Tübingen, 72076 Tübingen, Germany; PLindinger@t-online.de; 4Clinical Trials Unit UMG, University Medical Center Göttingen, 37075 Göttingen, Germany; rb@bredenkamp.com; 5Department of Sport Science, Institute for Nutrition, Sports and Health, University of Vienna, 1150 Vienna, Austria; 6Department of Nutritional Sciences, Institute for Nutrition, Sports and Health, University of Vienna, 1090 Vienna, Austria

**Keywords:** lifestyle intervention, health program, web-based, physical activity, physical fitness, overweight, obesity, weight loss, health effects

## Abstract

Web-based lifestyle interventions have attracted considerable research interest. Available evidence on such interventions suggests health-promoting effects, but further research is needed. Therefore, this study aims to investigate short-, medium-, and long-term health effects of a web-based health program (“TK-HealthCoach”, TK-HC) offered by a national statutory health insurance fund (Techniker Krankenkasse, TK). The study comprises two randomized controlled clinical trials to evaluate the health goals “Increasing Fitness” (F_clin_) and “Losing and Maintaining Weight” (W_clin_). A total of *n* = 186 physically inactive (F_clin_) and *n* = 150 overweight or obese (W_clin_) adults will be randomly assigned to a 12-week interactive (TK-HC) or non-interactive web-based health program using permuted block randomization with a 1:1 allocation ratio. Primary outcomes include cardiorespiratory fitness (F_clin_) and body weight (W_clin_). Secondary outcomes comprise musculoskeletal fitness (F_clin_), physical activity and dietary behavior, anthropometry, blood pressure, blood levels, and vascular health (F_clin_, W_clin_). All outcomes will be measured before and after the 12-week intervention and after a 6- and 12-month follow-up. Additionally, usage behavior data on the health programs will be assessed. Linear mixed models (LMMs) will be used for statistical analysis. Findings of this study will expand the available evidence on web-based lifestyle interventions.

## 1. Introduction

Today’s lifestyle in wealthy societies is increasingly characterized by physical inactivity and unhealthy diets, facilitating the prevalence of overweight and obesity. These lifestyle factors have been identified as global health risks in the latest Global Burden of Disease Study [1]. In 2019, 0.832 million deaths and 15.7 million disability-adjusted life-years (DALYs) were attributed to physical inactivity [2]. Unhealthy diets were blamed for 7.94 million deaths and 188 million DALYs [3], while overweight and obesity contributed to 5.02 million deaths and 160 million DALYs [4]. Particularly for noncommunicable diseases (NCDs), which are a challenging public health problem [5], these behavioral and metabolic risks are among the most modifiable and preventable risk factors [6].

In Germany, 79.5% of women and 75.3% of men are insufficiently physically active, i.e., they do less than 150 min of moderate aerobic activity and muscle-strengthening activities on less than two days per week [7]. They thus do not meet the recommendations for physical activity [8]. Considering current dietary guidelines [9,10], the German population consumes too much fat and protein, especially from animal sources, and too few fruits, vegetables, legumes, and fiber [11]. Moreover, 28.8% of women and 43.3% of men are overweight (body mass index (BMI) ≥ 25.0 kg/m^2^ [12]) and 18.0% of women and 18.3% of men are obese (BMI ≥ 30.0 kg/m^2^ [12]) [13]. Thus, almost half of German women (46.7%) and about two-thirds of German men (61.6%) live with increased body weight [13].

The consequences of these health risks, namely the increased mortality and morbidity risk of NCDs [14,15,16,17], can, however, be countered with the far-reaching potential of a healthy lifestyle promoting physical activity and a healthy diet. There is convincing evidence that these lifestyle factors contribute significantly to the prevention and treatment of many NCDs such as obesity, type 2 diabetes mellitus, and metabolic syndrome [18,19,20,21,22,23,24,25,26,27,28]. Furthermore, it has been shown that they induce positive effects on specific health outcomes associated with these diseases, including anthropometrics [29,30,31], blood pressure [32,33,34], blood glucose [31,35,36], blood lipids [31,36,37], and endothelial function [38,39,40]. Therefore, physical activity, a healthy diet, and normal body weight need to be addressed as essential components of lifestyle interventions for NCDs [41,42].

Most existing lifestyle intervention programs that focus on these components are face-to-face interventions. Their effectiveness in preventing and treating NCDs is already well-proven [43,44,45,46,47]. As a result of progressive digitalization, web-based lifestyle interventions have attracted much attention, mainly because of their accessibility regardless of location or time and comparatively lower costs [48]. Previous research on such interventions indicates a health-promoting potential [49,50,51,52,53,54,55,56,57,58,59]. However, there is still a significant need for further investigations, notably on long-term effects, successful intervention components, and objective health outcomes [60]. Regarding web-based interventions offered by national statutory health insurance funds, most of them refer to unimodal health programs [61,62,63,64] and have rarely been evaluated [65,66,67,68,69]. Based on this state of research, it is evident that web-based lifestyle interventions are still in their infancy and provide considerable research potential. This implies the development of preferably individually tailored, multimodal lifestyle intervention programs and their comprehensive evaluations.

The present study intends to close this gap by evaluating such a program of a national statutory health insurance fund (Techniker Krankenkasse, TK). This program (“TK-HealthCoach”, TK-HC) focuses on sustainable lifestyle changes to prevent NCDs in the long term by offering individually tailored, multimodal health coaching in which individuals can prioritize a personal health goal [70]. The evaluation of this program will be realized as a nationwide study by the scientific direction of the Section of Health Care Research and Rehabilitation Research (SEVERA) of the Medical Center of the University of Freiburg. Therefore, the TK-HC will be evaluated in three randomized controlled online trials (F_online_, W_online_, S_online_) investigating the effects on achieving the health goals “Increasing Fitness”, “Losing and Maintaining Weight”, and “Smoking Cessation” assessed by self-reported outcomes via online questionnaires. The study protocol of these online trials can be found elsewhere [71]. Concerning the health goals “Increasing Fitness” and “Losing and Maintaining Weight”, two additional randomized controlled clinical trials (F_clin_, W_clin_) will be performed to strengthen the effects on health goal achievement by objectively measured health outcomes, such as cardiorespiratory fitness (F_clin_) and body weight (W_clin_), through sports and nutritional medical examinations. For feasibility reasons, these two clinical trials will be conducted only in one specific geographic region of Germany, namely in the southwest of Germany, and will be coordinated by the Department of Sport and Sport Science (DoSS) of the University of Freiburg. This paper outlines the study protocol of these two clinical trials with adherence to the Standard Protocol Items: Recommendations for Interventional Trials (SPIRIT) statement [72,73].

The overall aim of the clinical trials (F_clin_, W_clin_) is to investigate short-, medium-, and long-term health effects of a 12-week interactive web-based health program (TK-HC) compared to a 12-week non-interactive web-based health program in physically inactive (F_clin_) and overweight or obese adults (W_clin_). For this purpose, the effects of each web-based health program on primary and secondary outcomes will first be examined. The hypothesis that participation in each health program is associated with improvements will be tested. Second, the difference between the interactive and non-interactive health program will be examined in terms of their effects on primary and secondary outcomes. The hypothesis will be tested that participation in the interactive health program achieves more significant improvements than participation in the non-interactive one. Finally, the impact of program use on program effectiveness will be examined. The hypothesis that higher frequency and intensity of program use leads to better primary and secondary outcomes in comparison to program use of lower frequency and intensity will be tested.

## 2. Materials and Methods: Design, Participants, Interventions, and Outcomes

### 2.1. Study Design, Setting, and Dates

The study is designed in the form of two randomized controlled, parallel-group clinical trials (F_clin_, W_clin_) using permuted block randomization with a 1:1 allocation ratio. As part of nationwide online trials [71], study participants (F_online_, W_online_) from the southwest of Germany (postcode area: 79) will be invited to medical examinations at the DoSS in addition to the online survey. These medical examinations will be carried out before (t0) and after (t1) the 12-week intervention and after a 6- (t2) and 12-month (t3) follow-up. The study is registered at the World Health Organization-approved German Clinical Trials Register (DRKS) (DRKS00020249, date of registration: 11 December 2019).

All interventions will take place online. The health programs were developed as desktop and mobile versions by TK and Vilua Healthcare GmbH (VHG), the project’s IT company. Thus, they will be accessible on electronic devices from anywhere at any time.

The evaluation project was planned and prepared from June 2018 to December 2019, including the implementation of two randomized controlled clinical pilot trials. Data collection started in January 2020 and will be completed by spring 2022.

### 2.2. Eligibility Criteria

For initial trial registration (F_clin_, W_clin_), subjects have to participate in the respective online trial (F_online_, W_online_) and have to live in the southwest of Germany (postcode area: 79). For trial inclusion (F_clin_, W_clin_), male, female, and diverse subjects must be between 18 and 65 years old, predominantly healthy, and willing to be randomized. In case of known health problems or diseases, individuals have to provide a medical certificate from a general practitioner attesting to their ability to participate in the trial and the health program. Specific inclusion criteria for the clinical fitness trial (F_clin_) are defined as a BMI of 18.5–34.9 kg/m^2^ and a physical activity level of ≤60 min/week. Pregnancy is set as an exclusion criterion. For the clinical weight loss trial (W_clin_), subjects with a BMI of 27.5–34.9 kg/m^2^ will be included, and women who are pregnant or breastfeeding will be excluded. Health problems or diseases that partially or entirely exclude the ability to participate in the trial or the health program are exclusion criteria of both clinical trials.

### 2.3. Interventions

#### 2.3.1. Interactive Web-Based Health Program

The intervention group for both clinical trials (F_clin_, W_clin_) will receive the TK-HC. This multimodal health program comprises different coaching modules to enable personalized health coaching according to the chosen health goal and personal health profile. With an evidence-based foundation, it attempts to meet the requirements of quality-assured prevention programs while providing a high degree of usability through interactive, self-directed, and flexible handling [70]. For the present evaluation of the TK-HC, however, the focus will be on individual coaching modules so that the intervention group of the fitness trial (F_clin_) will receive the “TK-FitnessCoaching” (TK-FC) and the intervention group of the weight loss trial (W_clin_) will get the “TK-WeightLossCoaching” (TK-WC). Both coaching modules are intended to achieve sustainable health effects within a 12-week intervention phase to prevent NCDs in the long term. The TK-FC is specialized in promoting physical activity and physical fitness based on endurance, strength, flexibility, and coordination training. The focus of the TK-WC is to assist users in losing and maintaining weight by following the energy density concept and a balanced, healthy diet.

The TK-HC is structured in the same way for both modules. Before starting the program, the user is guided through a multistep anamnesis to check suitability for the chosen health goal and set up an individually tailored program according to the personal health profile. The TK-WC user selects the preferred weight loss goal (3 or 5 kg). After completing the anamnesis, the user is directed to the program platform, consisting of three main areas. The first area (“My health program”) presents a personal dashboard that forms the central part of the program and accompanies the user in everyday life (Figure 1a,b). Here, the user can view an individual weekly plan of daily activities and an overall 12-week coaching plan based on three phases (phase 1: getting to know and trying out (week 1–3), phase 2: consolidating new behaviors (week 4–6), phase 3: anchoring habits (week 7–12)). The user can flexibly adapt these plans to individual needs by selecting activities from 13 fitness (TK-FC) (Figure 2a) and 12 weight loss (TK-WC) (Figure 2b) activities.

The fitness activities focus strongly on endurance training (e.g., “I improve my endurance by running”) and strength training (e.g., “I strengthen my muscles according to a recommended training plan”) as well as on flexibility and coordination training (e.g., “I train my flexibility”) and other health-related fitness courses (e.g., “I train yoga according to instructions”, “I train my back”). All training sessions follow the principle “train—log—analyze” and include detailed training introductions and instructions. Depending on the activity, the sessions can be done at home, in the gym, or outdoors. For endurance and strength activities, a standardized fitness test must be completed to classify the training level and thus select an adequate training plan. After six and eleven weeks, it is recommended to repeat this fitness test to measure the training success objectively. Besides all these activities, additional sports activities can be logged.

The weight loss activities concentrate on tasks that support the user in implementing the energy density concept (e.g., “I achieve a green energy density”) and the principles of a balanced, healthy diet (e.g., “I eat fruit two times and vegetables three times a day”). The key activity of the weight loss program is the nutritional protocol, which is automatically scheduled daily into the coaching plan. This tool supports adherence to the energy density concept and orientation on energy balance by displaying energy density, calorie intake, calorie consumption by physical activity, and fluid intake. In addition, this activity includes a daily macronutrient analysis and a cookbook with many healthy recipes. Another pre-planned activity is to log body weight and waist circumference every week.

All fitness and weight loss activities must be logged daily or weekly and can be modified at any time. The TK-FC user can also select weight loss activities, and the TK-WC user can also choose fitness activities. Both users can further link an activity tracker to the program by transferring the daily step count. Overall, the dashboard visualizes daily planned and logged activities, total activities completed per week, current health goal achievement via widgets, notifications to communicate with the user, personalized daily tips, and a support option. The TK-FC also includes a weekly barrier management tool.

In the second area of the program (“Knowledge”), the user receives evidence-based background information on the respective health goal through a coaching-accompanying knowledge course (“Optimal training” (TK-FC), “Weight loss with energy density concept” (TK-WC)), which is pre-set as a weekly activity in the coaching plan. Moreover, the user can pass other health-related knowledge courses. All courses are split into small articles and are structured interactively using texts, videos, tools, and tests.

Finally, the third area (“My data”) provides an individual analysis profile with visualized data from logged activities as well as the personal health profile and personal data. Furthermore, the user can download the study information and consent form for both the online and the clinical trial and has the option to withdraw participation from the trials and the health program. After program completion, the user gets a coaching summary, and the user can repeat the coaching as often as desired until the end of the study period.

#### 2.3.2. Non-Interactive Web-Based Health Program

The program for the control group for both clinical trials (F_clin_, W_clin_) will consist of a simply designed online platform that offers support in increasing physical activity and physical fitness (F_clin_) or losing and maintaining weight (W_clin_) over a 12-week intervention phase. In terms of content, it consists of a health-related knowledge transfer. Thus, only scientific background information on the respective health goal can be found, structured in short articles, but without interactive components. In addition, the user has access to personal data and the study information and consent form for the online and the clinical trial and has the option to withdraw participation from the trials and the health program.

The study participants of the clinical trials (F_clin_, W_clin_) will receive the respective health program after successful baseline assessment. They will log into the program via the project’s landing page. Criteria for discontinuing, interrupting, or modifying the allocated intervention are defined as feeling unwell, worsening health status, illnesses, diseases, injuries, harms, adverse events, or withdrawal from participation. These criteria will be determined by the study participant or the principal medical investigator of the study. Apart from the interventions, the participant will not be allowed to participate in other studies on physical activity or dietary behavior. For successful study participation and to promote adherence, the study participants will get free medical examinations and an activity tracker (Fitbit Charge 3^TM^) [74]. For participating in the online trial, they will obtain a shopping voucher and discount coupons. Apart from that, all study participants will be given free access to the current version of the TK-HC, including all coaching modules after study completion. As the TK-HC was frozen as a study version before the study started, the current version of the TK-HC contains some new content or features as it is continuously developed [71]. To maintain adherence during the study course, the study participants of the clinical trials will be reminded of the medical examinations by telephone and e-mail. To further monitor adherence, usage behavior data on the health programs will be analyzed within the formative evaluation of the online trials [71].

### 2.4. Outcomes

#### 2.4.1. Primary Outcomes

The primary outcomes aim to measure the achievement of the respective health goal. Therefore, the primary outcome of the fitness trial (F_clin_) is determined as physical fitness, especially cardiorespiratory fitness, measured by maximum oxygen uptake (VO_2_max) (ml/min/kg). Therefore, the study participants will complete the Cooper 12-min run test [75] on a 400 m running track. From the maximum distance covered in meters, the VO_2_max will be estimated using the formula “VO_2_max (ml/min/kg) = (distance in m − 504.9)/44.73” [76]. The test will be performed according to a modified standardized test protocol [77]. The heart rate (220 − age ± 10 bpm) and the Borg ratings of the perceived exertion (RPE) scale (≥17) [78] will be used as subjective and objective exertion criteria and assessed before, during, and after the test. The air temperature, humidity and pressure, and weather conditions will be documented. The study participants will complete a standardized 5-min warm-up and a 10-min cool-down program. Moreover, they will receive a standardized test meal [79]. In case of no exertion or stopping the test due to health problems, the study participants will have to repeat the test.

For the weight loss trial (W_clin_), body weight (kg) is defined as the primary outcome. It will be recorded in a standardized position in underwear and without any accessories (glasses, jewelry, watch) on a bioelectrical impedance analysis scale (seca mBCA 515) [80]. The study participants will be asked to use the toilet beforehand to empty their bladder.

#### 2.4.2. Secondary Outcomes

Various outcomes are set as secondary outcomes. First, for the fitness trial (F_clin_), a further criterion of physical fitness, especially musculoskeletal fitness, is maximum isometric strength (N, kg, lbs). It will be measured with an isometric leg press test using the ProFit Cube software [81] and a handgrip test using a Jamar hand dynamometer [82]. The measurements will be modified to standardized test protocols [83,84,85,86]. A standardized 5-min warm-up phase will precede both procedures. Concerning the first procedure, the study participants will take an individually standardized lying position in an isometric leg press [87] (backrest angle: 10°, slide position: individual, foot position (shoes off): 2, heel position: individual, shoulder fixation: resting against shoulders, arm position: crossed on chest, knee angle: 90°). After a submaximal familiarization phase (4 repetitions, 50–60% of maximum voluntary contraction (MVC): 2–3 s, rest: 1 min), the maximum strength (N) will be recorded bilaterally in three successful trials (3 repetitions, 100% MVC: 2–3 s, rest: 3 min). In the second procedure, the study participants will adopt a standardized sitting (sitting upright, upper arm: adducted to the body, elbow: 90° flexion, forearm: neutral position, wrist: 0–30° extension, ulnar deviation: 0–15°, other arm: relaxed next to the body) and grip position (second handle position). The maximum strength (kg, lbs) will be recorded in three successful trials, alternately left- and right-handed, by squeezing the handle of the hand dynamometer maximally for 2–3 s. The test will be started with the dominant hand. Jewelry and watches will be removed before the test. The maximum and mean value of three successful trials will be documented in both procedures. Verbal encouragement will be given in each trial.

Furthermore, secondary behavioral outcomes are defined for both clinical trials. Physical activity behavior will be measured in terms of steps taken (steps/day), sedentary behavior (minutes/day), and light, moderate, and vigorous intensity physical activity (minutes/week) using the activity tracker Fitbit Charge 3^TM^ [74,88,89,90] and the long version of the International Physical Activity Questionnaire (IPAQ-L) [91,92]. The activity tracker will be given to the study participants at the first medical examination. Data will be collected daily between the first two measurement time points (t0–t1) and seven days after the follow-up measurement time points (t2, t3), respectively, via a personalized study account. Dietary behavior will be assessed by energy and nutrient intake (kcal, µg, mg, g, %) and food consumption (Healthy Eating Index of the German National Nutrition Survey (HEI-NVS)) (g, ml, score) [93,94]. For the weight loss trial (W_clin_), energy density (kcal/g) will additionally be taken to check adherence to the energy density concept [95,96]. All outcomes will be analyzed based on nutritional protocols using the NutriGuide^® Plus^ software (Version 4.8) [97]. The nutritional protocols will be completed seven days after the first medical examination and seven days before the second, third, and fourth medical examination. The study participants will be introduced to the activity tracker and the nutritional protocol at the first medical examination.

Finally, secondary physiological outcomes are set for both clinical trials. Anthropometric outcomes include body weight (kg), body height (cm), BMI (kg/m^2^), fat mass and fat-free mass (kg, %), and waist circumference (cm). For the weight loss trial (W_clin_), however, body weight is considered the primary outcome. The above-mentioned outcomes will be assessed in a standardized position using a stadiometer (seca 274) [98] and a bioelectrical impedance analysis scale (seca mBCA 515) [80,99,100,101]. The waist circumference will be measured with a measuring tape (seca 201) [102] according to a standardized protocol at the midpoint between the lowest rib and the iliac crest. The study participants will be asked to keep their arms loosely at the sides of the body and place their feet together. The investigator will stand on the right side of the subjects. The measuring tape will be used parallel to the floor and lie flat on bare skin without pressing in. The measurement will be taken at the end of a normal expiration. The mean value of two successful measurements (difference: ≤1 cm) will be scored [103]. The study participants will be requested to use the toilet beforehand to empty their bladder, undress except for underwear, and remove jewelry, glasses, watches, and other accessories.

Moreover, systolic and diastolic blood pressure (mmHg) and heart rate (bpm) will be collected via a clinically validated and calibrated electronic blood pressure device [104]. The measurement will be taken on the left upper arm in a standardized way [105,106]. The study participants will be seated upright on a chair (back: supported, legs: uncrossed, feet: flat on the floor, left arm: supported and supinated, free of clothing, jewelry, and watch, right arm: relaxed on the thigh). The cuff will be applied at the right atrium level on bare skin using a correct cuff size (upper arm at least 80% enclosed). After a 5-min rest period, two measurements will be recorded at an interval of 1–2 min. The mean value of two successful readings (systolic blood pressure difference: ≤10 mmHg) will be taken for each parameter. Before and during the measurement, the study participants will be instructed to relax, not to move or talk. The measurement will be carried out in a closed and quiet room without background noise.

Further physiological outcomes are determined as blood levels, especially blood lipids (total cholesterol, low-density lipoprotein (LDL) cholesterol, high-density lipoprotein (HDL) cholesterol, triglycerides (mg/dl)) and blood glucose (fasting plasma glucose (mg/dl), glycosylated hemoglobin (HbA1c) (mmol/mol Hb, %)) taken by blood samples. The samples will be analyzed by the Medical Care Center (MVZ) Clotten in Freiburg.

Finally, vascular health will be examined by endothelial function as flow-mediated dilatation (FMD) (%) via the non-invasive AngioDefender™ system [107]. The assessment will be conducted on the left upper arm in a standardized procedure. The study participants will take the same sitting and arm position as for the blood pressure measurement. A correct cuff size will also be used, and the cuff will be applied on the upper arm at heart level accordingly. After a 10-min rest period, the test will take 8 min. Before starting the test, absolute and relative contraindications will be clarified. In case of pain, discomfort, or complications, the test will be stopped immediately. Before and during the test performance, the study participants will be instructed to relax, not to move or talk. The measurement will be performed in a closed, quiet, and temperature-controlled room (ideally 22–23° C) without background noise. With the FMD and other assessed lifestyle-related health outcomes, the vascular age (days) of the study participants will be calculated by using the Vascular Age Calculator (VAC) [108].

All primary and secondary outcomes will be assessed at four measurement time points (t0: 0 weeks, t1: 12 weeks, t2: 6 months after t1, t3: 12 months after t1), each taking about 4–5 h (F_clin_), and 2–3 h (W_clin_), at the DoSS. All medical examinations will be carried out at the same time of day using standardized procedures by trained and qualified sports and nutritional science and medical staff in separated rooms to protect privacy. The study participants will fast (meal and drinks, nicotine consumption: ≥12 h, alcohol consumption and exercise: ≥48 h). In addition, they will have to sleep sufficiently the night before and will have to be in a current state of good health. These criteria will be checked individually via an anamnesis at the beginning of each medical examination, in which current sensitivities, general habits and disorders of sensitivities, and medication, illnesses, and diseases will also be assessed. Moreover, in case of health problems or diseases, the medical certificate from a general practitioner attesting to the ability to participate in the trial and the health program will be checked, and adverse events will be documented. If these examination conditions are not met, or the examination cannot be performed as scheduled for other reasons, such as coronavirus infection, the examination will be rescheduled as soon as possible, depending on the individual’s state of health. Any rescheduled appointments will be carefully documented for consideration in the data analysis. Furthermore, at the beginning of each measurement, there will be a detailed introduction, instruction, and demonstration of the respective measurement procedure. In case of pain, discomfort, or complications, the measurement will be stopped immediately and, if necessary, repeated. An overview of all primary and secondary outcomes can be found in Table 1.

#### 2.4.3. Other Outcomes

To examine the effects of usage behavior on primary and secondary outcomes, system usage data regarding the participation in the health programs during the intervention phase (frequency (days with logins/week) and duration of program logins (average minutes/login), logged activities (activities/week) (intervention group)) and the follow-up phase (frequency of program logins (days with logins/week)) will be collected and stored by VHG. A secure cloud system will be used to transfer these data to the DoSS [71].

### 2.5. Participant Timeline

As the study participant flow of the clinical trials (F_clin_, W_clin_) combined with the online trials (F_online_, W_online_) is complex, its illustration is divided into the enrolment process (Figure 3a) and the intervention and assessment process (Figure 3b). Further details of the study participant flow of the online trials are described in another study protocol [71].

### 2.6. Sample Size

The sample sizes of the clinical trials (F_clin_, W_clin_) were calculated with regard to the long-term effect (t0–t3) of the respective primary outcome (F_clin_: VO_2_max, W_clin_: body weight) using G*Power software (Version 3.1) (Heinrich-Heine-Universität Düsseldorf, Düsseldorf, Germany) [109]. Based on comparable studies [50,110,111], effect sizes of Cohen’s *d* = 0.45 (F_clin_) and *d* = 0.50 (W_clin_), as well as a significance level of 0.05 and a statistical power of 0.80 were assumed. For the fitness trial (F_clin_), this resulted in a required case number of *n* = 79 for both the intervention and control groups. For the weight loss trial (W_clin_), the required case number was *n* = 64 for both study groups. Allowing for an additional dropout rate of 15% in each trial, the total target sample size was found to be *n* = 186 (93 + 93) (F_clin_) and *n* = 150 (75 +75) (W_clin_). Due to limited resources, there are no plans to exceed these sample sizes. Details can be found in the study protocol of the online trials [71].

### 2.7. Recruitment

The recruitment period for the clinical trials (F_clin_, W_clin_) is planned from January to June 2020. To achieve the target sample sizes, extensive recruitment measures will be carried out in the southwest of Germany (postcode area: 79) using a wide range of print and online media. These will mainly include local press and radio campaigns, flyers displayed in regional shops and centers, and Google advertisements. In addition, the institutions involved in the project will draw attention to the trials on their websites, newsletters, and social media. The recruitment process will be monitored to initiate appropriate measures if the required sample sizes are not reached by the planned end of the recruitment.

From recruitment to successful study enrolment, subjects will have to pass through a multistep process (Figure 3a). If subjects become aware of the study, they will be guided to the project’s landing page, first selecting their personal health goal. Afterwards, they will receive the study information for the respective online trial (F_online_, W_online_), provide written consent, and thus successfully register for the online trial. The study participants will then obtain the first online questionnaire by e-mail. After completing the questionnaire, the study participants from the postcode area 79 will be invited for the respective clinical trial (F_clin,_ W_clin_). To register for the clinical trial, they will have to agree to the study information (Appendix A) by their written consent form (Appendix A) and enter their contact details and time of availability. After successful registration, they will be randomized in the clinical trial. Within a few days, the study staff will contact the study participants by telephone or e-mail to provide more detailed study information, conduct a complete screening according to the inclusion and exclusion criteria, and make an appointment for the first medical examination. If the criteria are not met, the subjects will not be admitted to the medical examination. Still, they will remain randomized within the clinical trial and participate in the online trial. Within the first on-site medical examination, the eligibility criteria will be rechecked, and any outstanding questions will be answered. After a 7-day follow-up measurement of physical activity and dietary behavior, the baseline assessment and thus the study enrolment will be completed.

## 3. Materials and Methods: Assignment of Interventions

### 3.1. Allocation

To achieve study groups of approximately the same size in each clinical trial (F_clin_, W_clin_), the study participants will be randomly assigned to the intervention and control group with a 1:1 allocation ratio by permuted block randomization with variable block sizes of 4, 6, and 8. The generation of the allocation sequences was done by SEVERA using the RITA software (Version 1.50) (Universität zu Lübeck, Lübeck, Germany) [112]. The randomization lists were sent to VHG via a secure cloud system and are securely stored there. This will ensure allocation concealment so that the randomization for each clinical trial will be implemented without any influence. If the study participants have successfully registered for the clinical trials on the project’s landing page, the randomization for each trial will be carried out in an automatic computer-based way according to the generated lists [71].

### 3.2. Blinding

As the study participants of the clinical trials (F_clin_, W_clin_) will be informed about the interactive and non-interactive web-based health program, they will recognize which program they have been assigned to immediately after logging in. Thus, blinding the participants will not be possible. However, the healthcare providers and outcome assessors will be blinded, as they will only be informed about group allocation after all outcome assessments have been completed. Although the study participants will be asked not to mention their group affiliation during the on-site examinations, it is possible that the healthcare providers and outcome assessors will become aware of it. The data analysts of the clinical trials will not be blinded, as data will possibly reveal group membership [71].

## 4. Materials and Methods: Data Collection, Management, and Analysis

### 4.1. Data Collection Methods

Data of the clinical trials (F_clin_, W_clin_) will be collected on about 336 subjects at four measurement time points (t0, t1, t2, t3) at the DoSS. The feasibility of the data collection methods has been tested in pilot trials. The data will be collected in separated and protected premises by qualified staff using reliable and validated instruments with standardized procedures. The staff will be thoroughly instructed and trained on all measurements in a standardized manner. The data collection will be done using a case report form for each study participant. The study staff will carefully complete this form during the medical examinations, and the study director will subsequently review it to ensure high data quality. All data will be digitized and subjected to final plausibility checks. The system usage data collection methods are described in the study protocol of the online trials [71].

Various measures are planned to promote participant retention and complete follow-up to make data sets as complete as possible. First, for successful trial participation, the study participants will receive free medical examinations, an activity tracker (Fitbit Charge 3^TM^), a shopping voucher, discount vouchers, and free access to the current version of the TK-HC. Moreover, the participants will be reminded of each medical examination, and all examinations will be scheduled in advance. Furthermore, participant retention and usage behavior data will be monitored. The reasons for non-retention and non-adherence will be documented for data analysis and interpretation. If subjects do not adhere to the intervention protocols, all outcome data will still be assessed. For data analysis, however, non-adherent participants will be considered by including the usage behavior data.

### 4.2. Data Management

Clinical trial data (F_clin_, W_clin_) will be collected and stored in pseudonymized form. The allocation of person and pseudonymized code will only be possible via an allocation list, stored securely and separately from the coded research data. The research data collected by the case report forms will also be kept secure. The electronic data will be stored on password-protected computers, and backup copies will be made regularly and stored securely. The allocation list and the coded research data will only be accessible for the study director and the study staff, and they will be obliged by signature to maintain confidentiality about the data collection of the clinical trials. The system usage data will be collected and stored pseudonymously by VHG [71].

The DoSS will only transmit the pseudonymized research data to SEVERA via a secure cloud system for combined data analysis of the online and the clinical trials. VHG will transmit the pseudonymized system usage data to SEVERA and the DoSS. No data will be shared with any third parties. SEVERA and the DoSS will carry out the data analysis of the clinical trials. The results of the clinical trials will only be presented and published in aggregated form so that no conclusions can be drawn about individuals. All personal data of the clinical trials will be deleted three years after study completion. Finally, all clinical trial research data will be deleted after ten years.

All personal data of the clinical trials will be stored securely and separately from the research data. Personal data will not be published or passed on to third parties. The personal data entered on the project’s landing page for trial registration will be accessed by the study director via a back-office system of VHG. The study participants will agree to this by their written consent form. After completion of study enrolment, the back-office system will be immediately shut down, and the DoSS will securely store the data. Personal data of excluded subjects will be deleted immediately during the screening process.

Participation in the clinical trials and the health programs will be voluntary. The study participants will be able to withdraw participation at any time without giving reasons or incurring any disadvantages. They will be able to withdraw on their own via the health program or by informing VHG or the DoSS. In addition, the principal medical investigator of the study will be able to terminate participation prematurely if health impairments occur that partially or entirely exclude the ability to participate in the trial or the health program. In such a case, the personal data will be deleted immediately. The research data already collected will be stored anonymously and included in analyses and publications. No further data will be collected. The same procedure will be applied if participants continue participating in the health program despite withdrawing from the trial.

The participants will be informed about all data protection concerns of the clinical trials and will provide written consent. A data protection concept following the European Union General Data Protection Regulation (GDPR-EU) was developed between all project partners (SEVERA, DoSS, VHG) for the evaluation project. This includes a Joint Controller Agreement (JCA) signed by all project partners and data protection officers [71].

The blood samples will be the only biomaterials of the clinical trials. The MVZ Clotten in Freiburg will analyze and properly dispose of the blood samples.

### 4.3. Statistical Methods

For all analyses, the IBM SPSS Statistics software (Version 28.0.0.0) (IBM Corporation, Armonk, NY, USA) [113] as well as R (Version 4.0.5) [114] and RStudio (Version 1.4.1106) [115] will be used. All primary and secondary outcomes will be assessed on a metric scale level at all measurement time points. Therefore, linear mixed models (LMMs) will be used for statistical analyses of all outcomes. In the repeated measured data, the time points will be nested within cases, so the cases will be the cluster variable. The time points will enter the model as a continuous variable from baseline (0) to the end of the study (3). Random effects for intercept will be estimated. Degrees of freedom will be calculated with Satterthwaite’s method implemented in a specific R package (lmerTest). The basic model will consist of a cross-level interaction and random effects for intercept and slopes.

Level 1:

$Outcome=\beta_{0j}+\beta_{1j}*time+\beta_{nj}*X_{nj}+r_{ij}$

Level 2:

$\beta_{0j}=\gamma_{00}+\gamma_{01}*group+\gamma_{0n}*Z_{0n}+u_{0j}$

$\beta_{1j}=\gamma_{10}+\gamma_{11}*group$ $+u_{1j}$

The cross-level interaction will be between time and the dichotomous group variable (intervention vs. control group). The fixed effect $\gamma_{10}$ will represent the average change over time, while the cross-level interaction $\gamma_{11}$ will represent the difference in change between both study groups. Possible covariates or confounding variables, such as age and gender, can enter the model at the placeholder $X_{nj}$ on level 1 or $Z_{0n}\;$ on level 2.

To analyze short- and medium-term effects, the metric scale time point variable will be changed to a factor variable. Pairwise comparisons will be used with correction for multiple testing to estimate differences between each measurement time point. We will test group differences on time points t1, t2, and t3 as between-group contrast. Within each group, we will compare baseline with program completion (t0 vs. t1) and baseline with follow-up (t0 vs. t2). Additionally, we will contrast program completion with follow-up (t1 vs. t2) and change between follow-up and the end of the study (t2 vs. t3). Moreover, there is no need to account for multiple testing for the two primary outcomes as the two clinical trials (F_clin_, W_clin_) are considered as two independent trials.

According to the intention-to-treat (ITT) principle, all randomized cases will be included. In addition, only cases that have followed the per-protocol (PP) principle will be analyzed. While for ITT analysis, missing values will be imputed, in PP (complete case) analysis, only study participants with no missing values for the corresponding outcome will be analyzed. The ITT and PP analysis will be compared. An assessment of completed measurement time points will be used to test whether study participants who have completed all medical examinations differ in the corresponding outcome. Multiple imputation will be used to impute missing data in the ITT analysis. Influx and outflux statistics will be applied to identify factors that will serve as auxiliary variables for the imputation model. Sensitivity analysis will be performed to compare imputed with not imputed data.

The usage behavior data will be analyzed between the intervention and control group and within the intervention group. The analysis between the study groups will be done descriptively. For the analysis within the intervention group, the outlined model will be modified. Therefore, the group variable will be omitted, and a variable that codes usage behavior will replace the group variable in the model on level 2.

## 5. Materials and Methods: Monitoring

### 5.1. Data Monitoring

A data monitoring committee (DMC) for the clinical trials (F_clin_, W_clin_) will be established for the recruitment and retention process as the outcomes are associated with low risks. This non-independent DMC will be composed of some of the study staff and the study directors of the involved institutions (SEVERA, DoSS, VHG). However, the trial sponsor (TK) will only be informed about the process and will have no access to any data. There will be no conflicts of interest. As part of the recruitment process, the DoSS will report the current recruitment data of the clinical trials at weekly DMC meetings and will inform the entire project team and the trial sponsor. The report will be done based on the registrations in the back-office system, the telephone screenings, and the medical examinations carried out on-site. If technical or other unforeseen problems arise during the recruitment process, these will be discussed in the DMC and resolved immediately in direct consultation. Based on the report, the entire project team will be able to adjust the recruitment measures and finally decide on recruitment completion. Afterwards, the DoSS will continue reporting at regular DMC meetings and informing the entire project team and the trial sponsor about examined study participants and study dropouts.

### 5.2. Harms

No particular risks are expected in the context of the clinical trials (F_clin_, W_clin_). In rare cases, skin irritation or infection of the puncture site can occur after blood uptake. Furthermore, muscle soreness can occur in the following days after physical fitness assessments or training sessions of the health programs. No other risks are expected.

Apart from that, the study participants will have to agree to the inclusion and exclusion criteria, undergo a detailed screening, and, in case of health problems or diseases, provide a medical certificate attesting their ability to participate in the trial and the health program. Moreover, the potential risks were reviewed by the Ethics Committee of the University of Freiburg and the principal medical investigator of the study.

Adverse and any other events will be documented in a standardized way at the medical examinations. The study participants will be asked to inform the study staff if any event occurs in the interim study phases. Adverse events will be reported to the principal medical investigator of the study and the entire project team. If any event occurs, the study participants will be able to cancel, interrupt, or repeat the medical examinations. They will also have the possibility to cancel, interrupt, or modify the health program. If necessary, immediate care and treatment measures will be initiated by the principal medical investigator. He will also be able to terminate participation prematurely if health problems occur that partially or entirely exclude participating in the trial and the health program. Travel accident insurance, including stays on-site, will be arranged for all study participants.

## 6. Conclusions

This study is part of a nationwide evaluation project. The aim of its two randomized controlled clinical trials (F_clin_, W_clin_) is to investigate short-, medium-, and long-term health effects of a 12-week interactive web-based health program (TK-HC) compared to a 12-week non-interactive web-based health program in physically inactive and overweight or obese adults. An elaborate assessment of objective outcomes will be used to assess health effects. As this study will be conducted during the coronavirus disease 2019 (COVID-19) pandemic, health effects could be affected due to restrictions in daily life. Finally, this study addresses the need for further research on web-based lifestyle interventions.

## Figures and Tables

**Figure 1 ijerph-19-01393-f001:**
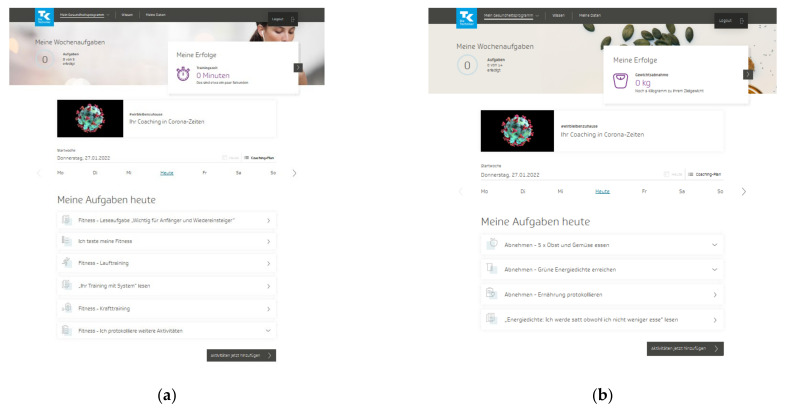
Dashboard of the “TK-HealthCoach” (TK-HC): (**a**) “TK-FitnessCoaching” (TK-FC); (**b**) “TK-WeightLossCoaching” (TK-WC). Source: Project’s landing page (https://gesundheitsprogramm.tk.de/de/studien/gesundheitsziele (accessed on 21 October 2021)).

**Figure 2 ijerph-19-01393-f002:**
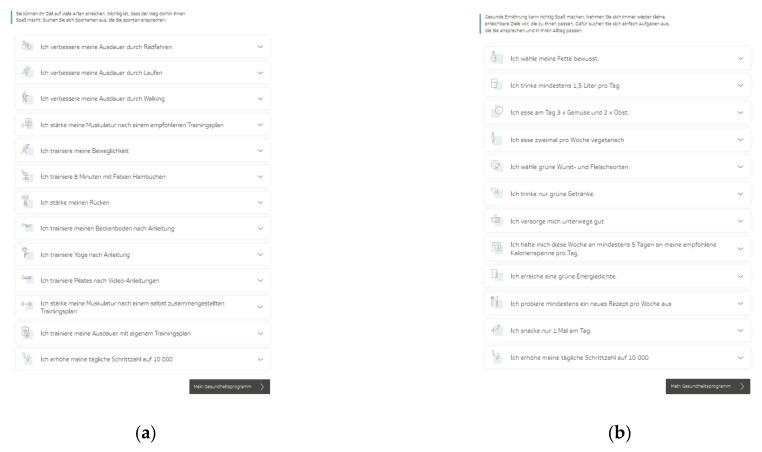
Activities of the “TK-HealthCoach” (TK-HC): (**a**) Fitness activities (“TK-FitnessCoaching” (TK-FC)); (**b**) weight loss activities (“TK-WeightLossCoaching” (TK-WC)). Source: Project’s landing page (https://gesundheitsprogramm.tk.de/de/studien/gesundheitsziele (accessed on 21 October 2021)).

**Figure 3 ijerph-19-01393-f003:**
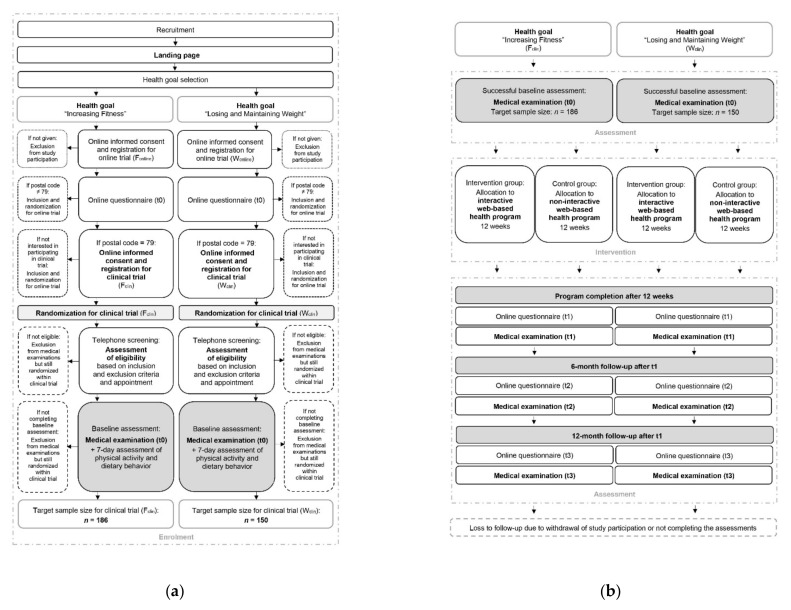
Participant timeline of the clinical fitness trial (F_clin_) and the clinical weight loss trial (W_clin_): (**a**) Enrolment process; (**b**) intervention and assessment process.

**Table 1 ijerph-19-01393-t001:** Primary and secondary outcomes of the clinical fitness trial (F_clin_) and the clinical weight loss trial (W_clin_).

Outcome						Measurement Time Point			
						t0 ^1^	t1 ^2^	t2 ^3^	t3 ^4^
primary	health goal achievement	F_clin_	cardiorespiratory fitness		maximum oxygen uptake (VO_2_max) (ml/min/kg)	x	x	x	x
		W_clin_	anthropometry		body weight (kg)	x	x	x	x
secondary	behavioral	F_clin_ W_clin_	physical activity behavior		steps taken (steps/day)	x	x	x	x
					sedentary behavior (minutes/day)				
					physical activity (light, moderate, vigorous intensity) (minutes/week)				
		F_clin_ W_clin_	dietary behavior		energy and nutrient intake (kcal, µg, mg, g, %)	x	x	x	x
					food consumption (HEI-NVS ^5^) (g, ml, score)				
		W_clin_			energy density (kcal/g)				
	physiological	F_clin_	musculoskeletal fitness		maximum isometric strength (N, kg, lbs)	x	x	x	x
		F_clin_	anthropometry		body weight (kg)	x	x	x	x
		F_clin_ W_clin_			body height (cm)				
					body mass index (BMI) (kg/m^2^)				
					fat mass (kg, %)				
					fat-free mass (kg, %)				
					waist circumference (cm)				
secondary	physiological	F_clin_ W_clin_	blood pressure		systolic blood pressure (mmHg)	x	x	x	x
					diastolic blood pressure (mmHg)				
					heart rate (bpm)				
		F_clin_ W_clin_	blood levels	blood lipids	total cholesterol (mg/dL)	x	x	x	x
					LDL ^6^ cholesterol (mg/dL)				
					HDL ^7^ cholesterol (mg/dL)				
					triglycerides (mg/dL)				
				blood glucose	fasting plasma glucose (mg/dL)	x	x	x	x
					HbA1c ^8^ (mol/mol Hb, %)				
		F_clin_ W_clin_	vascular health		endothelial function (FMD) ^9^ (%)	x	x	x	x
					vascular age (days)				

^1^ t0: 0 weeks; ^2^ t1: 12 weeks; ^3^ t2: 6 months after t1; ^4^ t3: 12 months after t1; ^5^ HEI-NVS: Healthy Eating Index of the German National Nutrition Survey; ^6^ LDL: Low-density lipoprotein; ^7^ HDL: High-density lipoprotein; ^8^ HbA1c: Glycosylated hemoglobin; ^9^ FMD: Flow-mediated dilatation.

## Data Availability

All data of the clinical trials (F_clin_, W_clin_) will remain the property of the DoSS. Only pseudonymized research data will be transmitted to SEVERA for data analysis. In addition, the results of the clinical trials will be presented to the scientific community and the trial sponsor as aggregated research data. Data on individual subjects cannot be reconstructed. No further data of the clinical trials will be released to the scientific community or third parties.

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
