# Peer review of "Health Effects of a 12-Week Web-Based Lifestyle Intervention for Physically Inactive and Overweight or Obese Adults: Study Protocol of Two Randomized Controlled Clinical Trials"

_ijerph, 2022, doi:10.3390/ijerph19031393_

Round 1
Reviewer 1 Report
Very interesting and very relevant study. However, it needs an extensive revision of the introduction as the hypotheses have not been adequately introduced.
In addition, please check the validity (and reliability) of methods and instruments selected for evaluation and interventions. Are the evaluations and interventions selected the best options to induce and track changes at every time point?
For me, power calculations are fantasy as every setting in each study is unique with contextual factors and interactions not considered in other contexts.
Minor
It is body mass, and not "body weight".
Reviewer 2 Report
The authors describe the study protocols for two randomized, controlled clinical trials aimed at investigating short-, medium-, and long-term health effects of a web-based health program on changes in physical activity and weight. Overall, the study is well-designed and the manuscript is well-written. The manuscript also adheres to the recommendations put forth in the SPIRIT guidelines for the reporting of protocols for intervention trials. I have no major comments or concerns. However, below I include a few minor points regarding the approach and analysis that may aid in improving clarity of the manuscript. My specific comments are provided below.
Abstract:
- Minor – The authors state that, “A total of at least n = 186 physically inactive (Fclin) and n = 150 overweight or obese (Wclin) adults will be randomly assigned…”. It is unclear under what scenarios the study would enroll more than n=186 and n=150 participants? If in fact the sample size targets are flexible, greater detail is required as to how the final sample sizes will be determined.
Introduction:
- Minor – In the last paragraph, the authors describe the goals of these trials, as well as those of the nationwide studies focused on self-reported outcomes. Is the main difference for these two trials that they will be looking at objectively measured outcomes? Whereas the others are looking at self-reported outcomes? If so, this point could be made clearer to readers as this is a major strength of these studies as compared to the others (and why being done in a specific geographic region due to feasibility etc.).
Methods:
- Minor – In section 1.2 it reads that, “First, for each trial, the effects of the web-based health programs on primary and secondary outcomes will be examined. The hypothesis that participation in each health program is associated with improvements will be tested. Second, the difference between the interactive and non-interactive health program will be examined in terms of their effects on primary and secondary outcomes. The hypothesis will be tested that participation in the interactive health program achieves more significant improvements than participation in the non-interactive one.”. I do not understand how these are different? Greater clarification is likely needed here.
- Minor – At line 361 and 362 of page 8 it is stated that participants, “…will have to sleep the night sufficiently before and will have to be in a current state of good health.”. How will this be operationalized and how will it be measured? What is to be done if these criteria are not meet? Are participants to be rescheduled? If so, might this put them out of window for the time point of interest? For example, given the state of things, what if someone is COVID positive at the time of a scheduled visit? How will this be handed? How much “margin”, in terms of days, are you allowing for a person to be considered to have been seen at a given time point?
- Minor – The sample size calculation does not appear to align with the primary analysis model as written. The sample size given looks to be based on testing for a standardized mean difference at a specific time point. The primary analysis model looks to focus on estimating change over time (with some mention of including time as factor in some models, etc.). Greater clarity might be achieved here my specifying at what time point this comparison it to be made (I assume 12 weeks). Also, greater detail should be given in how the formal testing will be carried out if using a mixed model since some approximation to the model denominator degrees of freedom (d.f.) is required to obtain p-values etc. or bootstrap resampling etc. Also, how exactly will the contrast will be applied to test for differences at specific time points using the mixed-model framework.
In addition, is there any need to account for multiple testing for the two primary outcomes? Perhaps not, given if these are viewed as two different trials, but some justification as to the rational is likely warranted.
- Minor – In section 4.2 the authors state that, “Possible covariates or confounding variables can enter the model at the placeholder ??? on level 1 or ?0? on level 2”. This seems an odd statement to me since, if we know the data generating mechanism (i.e., we know the subjects were randomized and how), what possible confounding covariates could there be? I would argue that confounding is not possible if the randomization is carried out properly and any imbalance you might see between groups is expected sampling variability which is accounted for in the analysis model. So, what would you adjust for here? More detail is needed, and it should be pre-specified.
Typically, one might adjust for prognostic factors though to explain part of the variably in the outcome since this may be expected to increase statistical precision. However, this should also be pre-specified.
- Minor – In section 4.2 it is stated that the multiple imputation will be used for the ITT analysis. I assume that the advantage of this approach over the MAR assumption inherent to the mixed model is the ease in which auxiliary variables can be included to extend the plausibility of the missing at random (MAR) assumption. How do the authors plan to identify factors that will serve as auxiliary variables? This should be pre-specified to the extent possible.
Also, it is stated that, “The missing data analysis will contain analyses of the frequency, patterns, and type of missing data (missing not at random, missing at random, missing completely at random). How will the authors identify patterns of missing not at random (MNAR) and MAR since these patterns cannot be ascertained from the observed data alone (i.e., we must make assumptions regarding these based essentially on expert knowledge)?
- Minor – For analyses looking at intervention “dose”, it is stated that, “The usage behavior data will be analyzed between the intervention and control group and within the intervention group. The analysis between the study groups will be done descriptively. For the analysis within the intervention group, the outlined model will be modified. Therefore, the group variable will be omitted, and a variable that codes usage behavior will replace the group variable in the model on level 2”. Did the authors consider dose as a mediator o the effect of the intervention on these outcomes? I might be missing something, but it seems that dose could be fitted as a mediating variable to obtain an estimate of the indirect effect of dose on the change in weight or vo2max by assigning all those in the control group a dose of zero (unless there is some equivalent for the control arm). This might be simpler, and directly address your question of interest.

Round 2
Reviewer 1 Report
There are no more comments. My reference to "body mass" was based on physics as mass and weight are different constructs. Congratulations.